# Survival of polymeric microstructures subjected to interrogatory touch

**Mickey Finn, III**☯, **Jeremy Treiber**☯, **Mahmoud Issa, Christian J. Martens, Colin P. Feeney, Lehna Ngwa, Charles Dhong, Darren J. Lipomi**(iD)*

Department of NanoEngineering, University of California, San Diego, California, United States of America

☯ These authors contributed equally to this work.
* dlipomi@eng.ucsd.edu

**Data Availability Statement:** Our data can be found in the Open Science Framework at DOI 10. 17605/OSF.IO/FVS2A (https://osf.io/fvs2a/).

**Funding:** This work was supported by the National Science Foundation under grant number CBET-

## Abstract

Polymeric arrays of microrelief structures have a range of potential applications. For example, to influence wettability, to act as biologically inspired adhesives, to resist biofouling, and to play a role in the "feel" of an object during tactile interaction. Here, we investigate the damage to micropillar arrays comprising pillars of different modulus, spacing, diameter, and aspect ratio due to the sliding of a silicone cast of a human finger. The goal is to determine the effect of these parameters on the types of damage observed, including adhesive failure and ploughing of material from the finger onto the array. Our experiments point to four principal conclusions [1]. Aspect ratio is the dominant parameter in determining survivability through its effect on the bending stiffness of micropillars [2]. All else equal, micropillars with larger diameter are less susceptible to breakage and collapse [3]. The spacing of pillars in the array largely determines which type of adhesive failure occurs in non-surviving arrays [4]. Elastic modulus plays an important role in survivability. Clear evidence of elastic recovery was seen in the more flexible polymer and this recovery led to more instances of pristine survivability where the stiffer polymer tended to ablate PDMS. We developed a simple model to describe the observed bending of micropillars, based on the quasi-static mechanics of beam-columns, that indicated they experience forces ranging from $10^{-4}$–$10^{-7}$ N to deflect into adhesive contact. Taken together, results obtained using our framework should inform design considerations for microstructures intended to be handled by human users.

## Introduction

### Background

High-aspect-ratio organic microstructures (e.g., posts, pillars, tubes, wires, and other shapes) occur naturally in the plant and animal kingdoms (e.g., lotus leaves and gecko feet). Moreover, they are potentially important for myriad technological applications, ranging from anti-fouling surfaces to materials with reconfigurable tactile properties, i.e., for haptic interfaces. In most envisioned applications, such structures will be subject to substantial mechanical insults, like contact by human fingers. Unlike high-aspect-ratio microstructures of biological origin, artificial ones cannot regenerate readily. Thus, it is necessary to develop criteria for the design of

1929748 awarded to DL. Additional support was provided by the Center for Wearable Sensors in the Jacobs School of Engineering at the University of California San Diego, and member companies Honda, Dexcom, Samsung, Huami, PepsiCo, Gore, Sony, Corning, and Merck KGaA in the form of financial support awarded to DL. This work was performed in part at the San Diego Nanotechnology Infrastructure (SDNI), a member of the National Nanotechnology Coordinated Infrastructure, which is supported by the National Science Foundation (Grant ECCS-1542148). None of this funding came in the form of salary or compensation to any author. The funders had no role in study design, data collection and analysis, decision to publish, or preparation of the manuscript.

**Competing interests:** Center for Wearable Sensors in the Jacobs School of Engineering at the University of California San Diego, and member companies Honda, Dexcom, Samsung, Huami, PepsiCo, Gore, Sony, Corning, and Merck KGaA in the form of financial support awarded to DL. There are no patents, products in development or marketed products associated with this research to declare.

structures such that they are likely to survive interrogatory touch by human users. Here, we fabricated a series of high-aspect-ratio micropillars in a photocurable resin which differed in interpillar spacing, height, diameter (and thus aspect ratio), and elastic modulus. We subjected these arrays to a tangential force provided by a silicone replica of a human finger. The downward pressure and lateral contact area was designed to approximate normal interaction of human fingers with real materials and devices—i.e., "interrogatory touch." The goal was to determine the design criteria necessary for micropillar survival in the laboratory using a setup that simulated a realistic scenario.

Since the development of soft lithography in the 1990s [1], high-aspect-ratio polymeric microstructures fabricated by replica molding in silicone templates have been central to a range of proposed applications. In many of these applications, the role of the microstructuring is to modify the interfacial forces [2–4]. Taking inspiration from structures in nature, superhydrophobic and oleophobic surfaces modeled after the lotus leaf have become an important area of research [5–8]. Similar structures are under investigation for their abilities to retard the formation of biofilms (e.g., of bacteria on medical devices or of barnacles on the hulls of ships) [9]. While these structures are meant to reduce adhesion, it is also possible to engineer other types of high-aspect-ratio structures for the purposes of promoting it. For example, the long, thin keratinous spatulae of the gecko have inspired the development of a range of bioinspired adhesives based on van der Waals forces [10–13]. The structures on the toes of tree frogs modify hydrodynamic forces that can permit the animal to adhere to surfaces, even under water [14]. In human-engineered applications, high-aspect-ratio relief structures are used for their mechanical behavior. For example, conical structures composed of biodegradable polymers and drugs have been used as microneedle patches for noninvasive drug delivery [15–17]. These structures are designed to puncture the skin and dissolve. In contrast to this application, contact with the skin of micropillars for most other applications is potentially highly damaging to the relief structures, yet almost impossible to avoid in everyday use. One potential application of polymeric microstructures is in haptics [18, 19], where contact with the skin is not just unavoidable but the very reason for these structures to exist [20]. Applications in robotic touch, [21–24] along with haptics-enhanced minimally invasive surgery [25] and virtual reality [26] require contact with human skin at realistic (i.e., significant) contact pressures [27].

Because of the importance of these high-aspect-ratio polymeric structures, along with the necessity to have them exposed to the environment, it would be highly beneficial to develop a framework for avoiding breakage and collapse of the structures when subjected to realistic forces by imperfect, highly non-ideal indenters (i.e., human fingertips exploring surfaces —"interrogatory touch") [28]. Our approach is thus different from most other studies on the failure of polymeric microstructures, which use idealized indenters or sliders in highly controlled conditions [29–31].

Degradation in function of an array of polymeric microstructures can occur by breakage of the structures, collapse of the structures, or deposition of material from the comparatively soft indenter (i.e. the finger) onto the array. These modes of degradation originate from intensive and extensive properties of the structures and of the array. Intensive properties deriving from the polymeric material include strength, toughness, modulus, and surface forces, while extensive properties derive from the geometries of the individual microstructures and of the array (e.g., spacing, diameter, and aspect ratio). Breakage occurs when the applied tangential force overcomes the ultimate tensile stress of the polymeric structures. In principle, it is straightforward to mitigate breakage by increasing the strength of the polymer (greater molecular weight, crosslink density, and intermolecular forces). Collapse occurs due to bending and subsequent adhesion to another microstructure (lateral collapse) or to the substrate (ground collapse) [28]. As a mode of failure, collapse is more difficult to mitigate than breakage, as all sufficiently thin

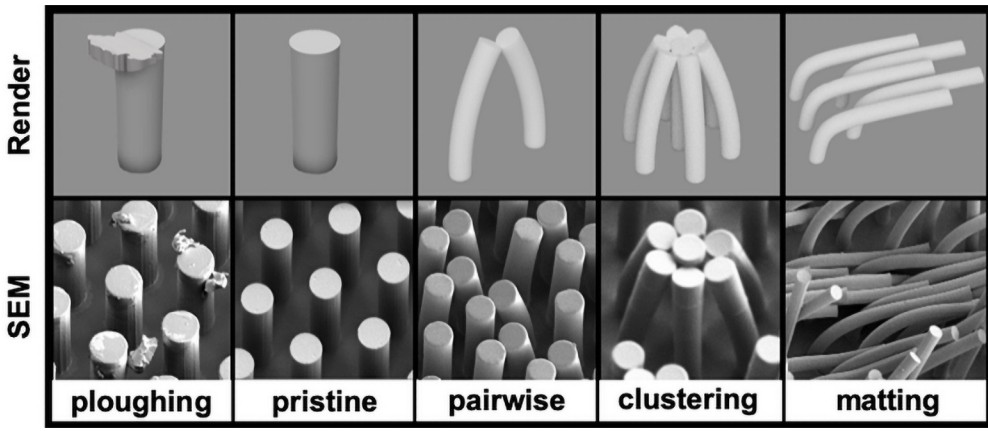

**Fig 1. Spectrum of mechanical responses.** Digital rendering and example SEM images showing a range of deformation outcomes in order of decreasing micropillar bending stiffness.

structures exhibit appreciable bending upon application of a tangential force, regardless of the modulus (i.e., low bending stiffness). There are at least three ways in which structures can collapse (Fig 1). For example, "pairwise collapse," in which two adjacent structures adhere at their tips, "clustering," in which a group of structures adhere at an apex, and "matting," in which groups of structures bend in the same direction.

## Theory

**Lateral collapse (pairwise and clustering).**   The tendency toward lateral collapse stems from a competition between the elastic energy stored in bent pillars and the adhesive energy between them [32, 33]. In the case of pairwise collapse and clustering, Glassmaker et al. formulated a critical aspect ratio beyond which patterned cylinders made of polydimethylsiloxane (PDMS, Sylgard 184) laterally collapse as a result of the strong adhesion between adjacent pillars once in contact

$$\frac{l}{d} = \left( \frac{0.57 E^{\frac{1}{3}} a^{\frac{1}{2}}}{\gamma_s^{\frac{1}{3}} d^{\frac{1}{6}} (1 - v^2)^{\frac{1}{12}}} \right) \tag{1}$$

where l/d is the aspect ratio of length over diameter, a is the pitch, $E$ is the elastic modulus, $\gamma_s$ the surface energy, and $v$ is the Poisson ratio of the material system [32, 34].

**Matting.**   In the case of matting, Persson derived a critical aspect ratio beyond which arrays of fibers would condense and form tilted compact layers. By considering the van der Waals interaction between fibers, $\varepsilon$, the curvature of the bent fibers, $\theta$, and the elastic modulus of the fibers, Persson concluded that elastic fibers would fail by matting when they exceeded the aspect ratio [33]:

$$\frac{l}{d} = \frac{1}{2} \left( \frac{\pi}{24} \frac{E d^2}{\varepsilon} \right)^{\frac{1}{2}} \theta \tag{2}$$

This model is especially interesting because there is a dependence on the curvature of the fibers leading to lateral collapse. Persson's derivation has an indirect dependence on the density of the array because of there is an increase in elastic restoring force with increasing fiber curvature. Tightly packed arrays may prevent failure by matting because the reduced penetration past the plane defined by the tops of the pillars can prevent a large enough angle from being obtained.

**Ground collapse.**    Ground collapse, a type of degradation especially likely in low-density arrays, has been predicted with derivations by Roca-Cusachs et al. [35]. The authors determined that the critical aspect ratio is:

$$\frac{l}{d} = \frac{\pi^{\frac{5}{3}}}{2^{\frac{11}{3}}3^{\frac{1}{2}}}(1 - v^2)^{-\frac{1}{6}}\left(\frac{E}{2\gamma_s}\right)^{\frac{2}{3}}d^{\frac{2}{3}} \tag{3}$$

assuming that the pillars and the substrate are formed of identical materials. Each of these models only considers elastic deformation held in place by adhesive forces alone; there are no considerations of external loads on the pillars or plastic deformation which is expected to lower the critical aspect ratio. Furthermore, these models do not consider buckling from gravitational forces as described by Hui et al. [36]. However, with increasing surface area to volume ratio for standing structures, the effects of gravitational forces are overcome by adhesive forces for structures with sizes in the micrometer to nanometer range [28].

The deflection of the pillars and, consequently, the extent of degradation of an array will depend to a large degree on the extent of adhesion and tangential force (i.e., friction) needed for the slider (or finger) to traverse the substrate. Amontons-Coulomb friction law state that the frictional force opposing sliding is proportional to the normal load and independent of the apparent area of contact [37]. In their seminal work on friction [38], Fuller and Tabor explained this law through the difference between the apparent versus real area of contact. They posited that the solids in contact are largely supported through the summits of surface irregularities such that intimate contact occurs over an area so small as to be independent of the size of the surfaces. With an increase in normal force, increasing numbers of asperities come into contact in order to accommodate the load and this results in intense local pressure leading to plastic deformation and flow. Fuller and Tabor observed that when one material is harder than the other and sufficient tangential force was applied to initiate sliding, the harder material tended to abrade the softer material in a phenomenon they refer to as "ploughing". Fuller and Tabor were also among the first to experimentally show how the interfacial adhesion between elastic solids decreases with increasing roughness of the substrate [39]. They further showed that adhesive effects due to surface roughness are more significant for interrogatory work pieces (i.e., the indenter, slider, or finger) with higher elastic moduli. Both relationships can be understood as the relationship between surface energy and contact area. Rough surfaces with large asperities reduce the contact area in the interface for high-elastic moduli materials [40], while softer materials can conform to the rough surfaces and increase the contact area [41].

Persson et al. were able to apply JKR theory—which describes adhesive force between a ball on a flat surface when one is elastic [41]—to formulate a quantitative model to calculate interfacial adhesion of a smooth rubber ball on a hard and rough surface as a function of surface roughness [42]. This idealized scenario involving a smooth slider surface does not account for surface irregularities on complex surfaces and cannot be directly applied to finger sliders.

Formulating a model to predict the degradation of a micropillar array that accurately describes the effects of the friction and adhesive forces involved when a finger makes dynamic contact with the micropillar surface remains a challenge. An attempt at modeling a similar interaction has been made by Degrandi-Contraires et al. [43]. However, in their case the interrogatory material was a smooth elastomer. No attempts have been made to understand this interaction with a realistic model of fingertip, which contains fingerprint ridges and other idiosyncratic asperities of a real finger that might serve to concentrate force on the array at various "hot spots" on the finger. To produce an accurate model based on empirical observation would require testing with either human subjects or automated mechanical actuation

involving a mock finger. Human subject testing is inherently time consuming and difficult to control owing to the variability of human fingers and the difficulty in controlling the level of moisture and the force applied [44, 45].

## Materials and methods

### Replica finger probe

Our approach is thus a hybrid between a tribology experiment which uses a slider with an idealized shape, and one which uses human participants. Namely, we used a silicone rubber cast of a human finger connected to an actuator programmed to deliver and monitor a constant load across an array of micropillars (Fig 2). Designing this realistic finger model was challenging because it required a model with mechanical properties and an epidermal morphology that is similar to an actual finger [46]. One particular challenge arose from the fingerprints. While these ridges may appear to be a loosely periodic train of wavelike crests to the naked eye, the curvature at the top of these ridges varies from ridge to ridge on the same finger and also varies between individuals. Furthermore, examination of fingerprint molds under magnification shows evidence of another order of roughness arising from irregular surface asperities and depressions—i.e., microrelief [47]. These structures fell within the order of 10 μm in all dimensions but were most commonly 5–30 μm ($mean = 12\ \mu m$, $\sigma = 7.7\ \mu m$, $n = 81$) with an apparent exponential distribution, according to image analysis on a SEM image of a representative sample.

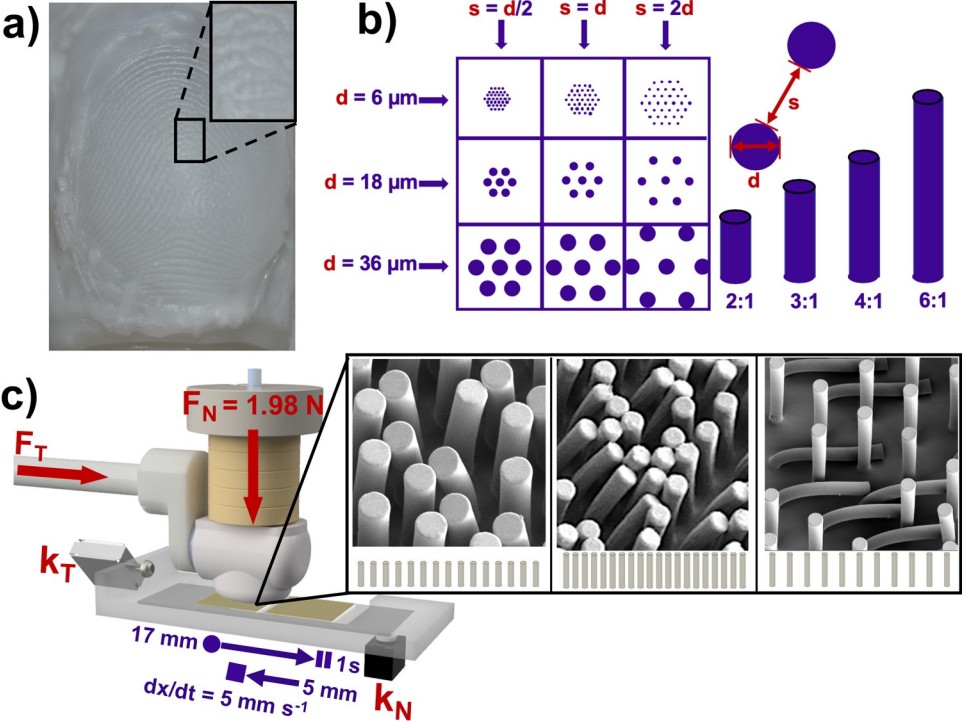

**Fig 2. Summary of the experimental design.** (a) Surface of PDMS replica finger with fingerprint ridges molded from human fingers. (b) Overview of geometrical parameters, namely micropillar diameter, aspect ratio and interpillar spacing, that were varied in this experiment. (c) Rendering of replica finger showing the sliding path across substrate carrier. The inset shows example SEM images of the phenomena we refer to as pair-wise, clustering, and matting. A complete schematic and photographs of the testing apparatus can be found in the S1 Fig.

In order to emulate this behavior with our replica finger probe, we used paraffin wax to take a mold (negative replica) of our fingerprints and cast the replica using PDMS (Sylgard-184, 30:1 base:crosslinker) that has an elastic modulus roughly approximating that of human skin [48]. Collection of participant samples (fingerprints) was done under the supervision of the University of California, San Diego Human Research Protections Program Institutional Review Board under project #191950S. Informed written consent was obtained for collection of participant samples. Upon release of the mold, we subjected the surface of the replica finger to a UV-ozone treatment to make the surface glassy and to reduce the tackiness [49, 50]. This material could not however, simulate the extent that lubricated sliding occurs on real fingers due to lubrication from sweat, sebum and other oils, nor does it approximate possible osmotic repulsion due to the electric double layer that forms on the skin in atmosphere [51]. The contribution and variability of these factors in real fingers when compared to a more simplified contact model such as our replica PDMS finger could be an issue to investigate further. Regardless, we assumed that the contact probe we used to simulate interrogatory touch could be modeled as an elliptical region of skin-compliant elastomer with roughness imparted by periodic arrays of ridges.

## Design of arrays

Our goal was to determine the effects of realistic human touch on the survivability of arrays of polymeric micropillars by varying (1) the geometry of the array and (2) the mechanical properties of the polymer. By subjecting epoxy micropillar arrays of varied elastic modulus, aspect ratio, diameter, and spacing to the shearing movement of a PDMS replica finger, we hypothesized that we could arrive at useful guidelines for survivability across a wide variety of materials. We chose hexagonally arrayed cylindrical micropillars as our representative relief structure due to their analytical utility and equidistant spacing. We chose cylindrical pillars because the corners of, e.g., a square column, might introduce unwanted anisotropy. The simplicity of the circular polar moment allowed for more compact calculations regardless of their orientation with respect to the direction the replica finger traveled across the array. Additionally, there is a wealth of literature involving the mechanics of cylindrical columns due to their use as structural elements throughout history. Hexagonal packing likewise assured that the orientation of the array with respect to the replica finger would have no bearing on its response. A square array, on the other hand, would be expected to have a different frictional response depending on whether the finger moves orthogonally or diagonally across the arrays due to the elements having a distance d vs d$\sqrt{2}$ separation between them.

The arrays were made using standard soft lithographic techniques, with masters made from silicon patterned with SU-8 negative photoresist as the relief structures (S1 Appendix). From these masters, PDMS molds were made. To fabricate our micropillar arrays, we used Norland Optical Adhesive (NOA), a line of photocurable resins which exhibit a wide range of mechanical properties [52–54]. We selected two products, NOA 73 ($E = 11$ $MPa$) and NOA 81 ($E = 1379$ $MPa$). We believed that the two orders of magnitude difference in elastic modulus would lead to an appreciable difference in the mechanical response of the micropillars. For convenience, we will refer to NOA 73 as the $E = 10$ $MPa$ material and NOA 81 as the E = 1000 MPa material going forward.

## Test motion of experimental probe

Our custom-built tribology apparatus (Fig 2 and S1 Fig) used slotted calibration masses to apply a 200 g (1.98 N) constant normal force. A stepper-driven linear actuator (ET-100, Newmark Systems) provided approximately 95 N of axial loading to move our replica finger across

the micropillar arrays. A resin-printed replica phalanx ("bone") was suspended in the PDMS mold of the fingertip and provided the means to mount it to a PTFE shaft with a diameter of ca. 10 mm. We initially mounted the replica on an aluminum shaft of similar size but found that metal was too stiff and supported the applied 200 g load so completely that the load failed to push the replica finger into greater contact with the micropillar arrays. A more elastic shaft made of PTFE, however, readily deflected under load but was stiff enough to allow the replica finger to be pushed laterally by the stepper motor. Each micropillar array comprised an area of 20 mm × 20 mm, so we decided on a testing surface in which we stamped NOA micropillars upon two adjacent areas to create an array surface that was 20 mm wide by 41 mm long with 1 mm separating the two stamped arrays. This arrangement allowed the mass loading to take place while the replica finger was centered and in contact with the surface of the first die, leaving enough horizontal distance for the replica finger to traverse outside of this initial contact area.

The micropillar arrays were stamped onto 2.5 cm × 7.5 cm glass microscope slides that were mounted in a milled acrylic carrier for testing. This carrier was supported at one end by a tangential force sensor (Futek LSB200 S-beam load cell, 2.5 N capacity) that was mounted to the fixed end of the linear actuator. Micrometer-actuated positioning stages were used to bring both the replica finger as well as the normal force sensor (Honeywell FSG005WNPB, 5 N capacity) into gentle contact with the top and bottom of the test substrate, respectively. After a baselining procedure to ensure that the test surface was level and under approximately the same loading for all samples, the linear actuator drove the replica finger at a velocity of 5 mm s$^{-1}$ in the forward direction for 17 mm, at which time it would pause for 1 s before moving in the reverse direction for 5 mm. We chose a forward- then backward-traveling test routine because, in the course of experimentally moving our fingers across a mass balance, we observed the spikes in normal force that occur when arresting motion or changing direction (possibly a signal of damage to the micropillar array). These movements seemed natural for human users in the course of handling objects or exploring surfaces.

## Results and discussion

### Effect of interrogatory touch on micropillar arrays

As intended, our selection of material properties and array geometries allowed us to observe the whole range of predicted behavior: lateral collapse, ground collapse, ploughing of the "finger," and pristine survival. The general trend of survivability vs. deformation that we observed was dominated by the aspect ratio of the micropillars. As shown in Fig 3A, both the 6 μm and the 18 μm (Fig 3C and 3D) micropillars experienced irrecoverable deformation at the 6:1 aspect ratio. The greater compliance of the 6 μm micropillars, however, is clear by their lateral collapse at the 4:1 aspect ratio as well. Taken together, these failure modes among the 6 μm and 18 μm micropillars, regardless of elastic modulus, underscore how much the slenderness of relief structures can modify their stiffness. Our results indicate that this effect is further mediated by the diameter of these structures.

We must call attention to the fact that the plots shown in Fig 3 are strictly qualitative; they were constructed by comparing scanning microscope images taken at multiple points along the replica fingers path of travel. These images were taken first at extremely low magnification to determine the extent (or lack) of deformation and then zoomed in to capture clear representations of that behavior for analysis and categorization. Our survivability criterion was based loosely on these microfabricated surfaces remaining functional after this experimental mimic of interrogatory touch and we presumed this should amount to the microcontact interface being preserved. The ploughing condition, therefore, was considered to constitute survival

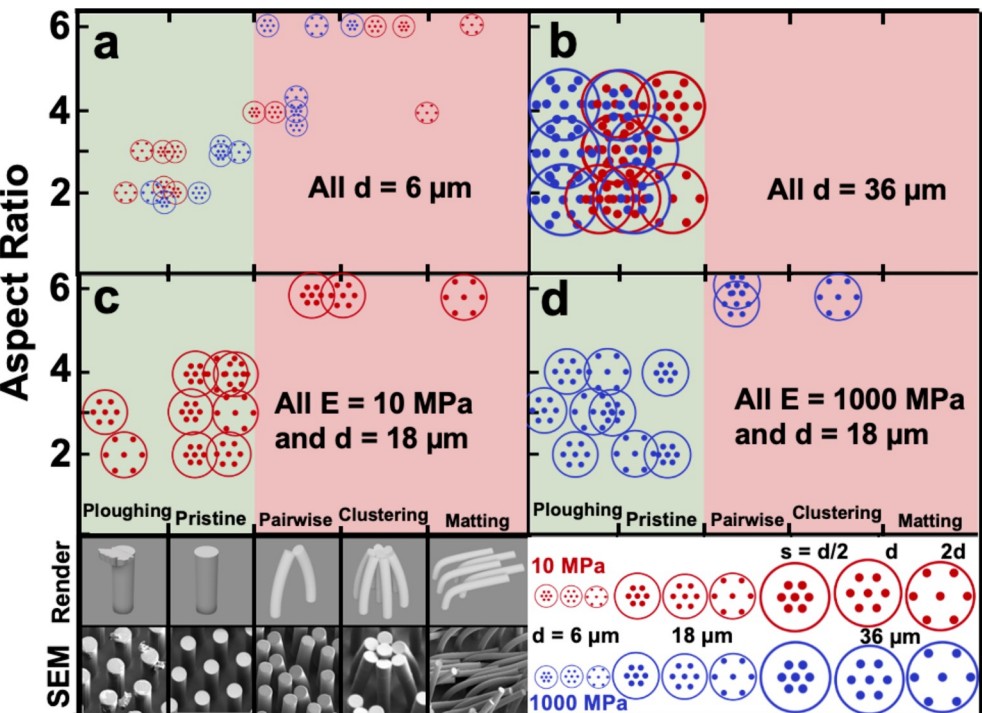

**Fig 3. Survivability of tested micropillar arrays.** Panel (a) displays the smallest diameter micropillars (d = 6 μm) that were the most prone to deformation and the only diameter to display lateral and ground collapse phenomena at both 4:1 and 6:1 aspect ratios. In contrast, panel (b) shows the outcome for micropillar arrays with the largest diameter (d = 36 μm) for both materials. The medium diameter (d = 18 μm) micropillars for the more compliant material (E = 10 MPa) are shown in (c) while the medium diameter micropillars composed of the stiffer polymer (E = 10 MPa) are shown in subpanel (d).

while the instances of lateral collapse and ground collapse failed simply because micropillars thus deformed can lead to a change in contact area, texture, wettability, etc. As far as the occurrence or yield of deformed micropillars required to categorize their survivability, we originally established a criterion that a deformed micropillar was an outlier if there were no similar defects within ten micropillars in any direction. This criterion was seldom used in practice, however, because we observed that deformed micropillars (or evidence of ploughing) tended to be either grouped much closer together or much farther apart. We also wanted our determination of survivability to be robust so we chose to be strict in our assessment of survivability such that failure anywhere in the path of travel was judged as failure for the entire array. Our reasoning for this strict criterion was that we did not want to qualify a given array in an unrealistic fashion such as stipulating that it can survive interrogatory touch provided that it is only touched once without stopping or changing directions. However, the initial contact area was an exception to this criterion due to idiosyncratic manipulations in setting up each sample for testing. That is, the length of time it took to carefully load the 200 g of calibration mass and fine-tune the apparatus to a repeatable baseline led to geometric aging of the contact due to the viscoelastic creep of the PDMS replica finger. Such creep increases the real area of contact of the interface beyond what the remaining travel path experiences, leading to an increased frictional response in that region.

The micropillar states shown in the horizontal axis of Fig 3, ranging from ploughing (of the PDMS replica finger) to matting (of micropillars due to lateral and/or ground collapse), encompassed all phenomena we observed and we thought if appropriate to order them left to

right by increasing flexibility of the micropillars. Subtle differences in horizontal placement are intended to convey differences in severity and/or extent of deformation and were especially helpful in recognizing trends in the effect of micropillar spacing within the arrays.

The effect of spacing is especially evident when comparing micropillar arrays that are too stiff to be deflected by the replica finger, highlighting the importance of how the competition in stiffness between two drastically different material surfaces, in geometry and mechanical properties, leads to a micropillar's survivability or mechanism of failure. Examining first the higher modulus material with the largest diameter (E = 1000 MPa, d = 36 μm, Fig 3B, shown in blue), we conclude that these parameters both point towards inflexibility of the micropillars regardless of aspect ratio. By comparison, the micropillars composed of the more flexible material generally ploughed less PDMS–a possible indication of deflection and elastic recovery. The stiffer micropillar arrays, however, all appeared to plough PDMS equally except for those with the tightest interpillar spacing (s = 1/2d = 18 μm). This is presumably because these arrays most closely resembled a flat surface and the distance between micropillars was small enough that the fingerprint ridges were afforded less purchase, resulting in the replica finger sliding across the tops of these structures with significantly less mechanical insult.

We observed an interesting trend among arrays with the same aspect ratio and material properties but with micropillars half the diameter of those previous (d = 18 μm, Fig 3D). At each of the three shortest aspect ratios, the three different micropillar spacings each exhibited distinguishable behavior but with the same trend in horizontal ordering. As with the case of the d = 36 μm micropillars, the arrays with the tightest interpillar spacing were the most pristine. This outcome is possibly the result of a greater top contact area resulting in a lower concentration of stress on any individual micropillar and, overall, less damage to the PDMS "skin." If much of the replica finger was in contact with the micropillar arrays, however, then the arrays with the widest spacing should plough the most PDMS because those micropillars experience the greatest stress locally. This is not the case, however, as Fig 3D shows that the arrays of intermediate spacing alone are capable of severely ploughing the PDMS skin among the arrays with 18 μm diameter micropillars.

This apparent discrepancy illustrates how the geometric and material properties combined to determine mechanical response in our experiment. It can be seen in Fig 3B that the more flexible polymer ablated less PDMS in all cases for the 36 μm diameter micropillars, implying that some deflection took place. It is also likely that for all micropillars with the same diameter composed of the stiffer polymer, such deflection occurred only minimally. Except for those arrays with the smallest spacing, the severity of ploughing was indistinguishable. When the diameter was decreased to 18 μm (Fig 3D), however, the degree of ploughing became distinguishable. As the arrays increased in aspect ratio from 2:1 to 3:1, ploughing became markedly more severe (by visual inspection). We suspect that these longer micropillars were able to penetrate deeper into the replica finger; even the most tightly-spaced array abraded the PDMS finger somewhat. When the aspect ratio was increased to 4:1, however, the decrease in bending stiffness was enough that the micropillars were able to deflect slightly and so caused less damage to the replica finger. Among the pillars at these three aspect ratios, greater ploughing could simply be the result of greater micropillar density, until the spacing is tight enough that the replica fingerprint ridges do not penetrate between individual structures. We suspect that stress concentration also plays a role in the extent of ploughing, as the arrays with the least density (s = 2d) will deflect more and offer less resistance to the replica finger. The spacing of the micropillars thus affects the severity of PDMS ploughing, which we consider to be a state of survival for a microstructured surface and liken it to the way human fingers slough away skin cells when drawn across a rough surface.

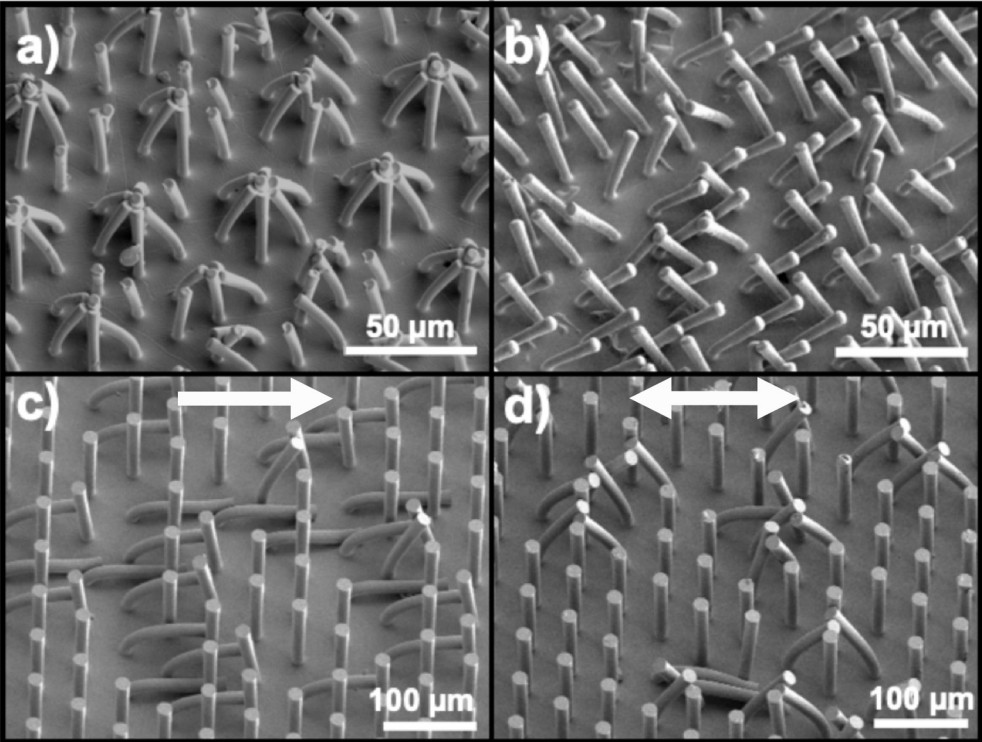

**Fig 4. Representative SEM images showing the deformation of micropillar arrays.** Panel (a) shows an example of "clustering" lateral collapse by the 6 μm diameter elements for the compliant ($E = 10$ $MPa$) material. In contrast, panel (b) shows the stiffer material ($E = 1000$ $MPa$) with the same geometry exhibiting pairwise lateral collapse. Both images (a) and (b) were taken in the bidirectional region of the travel path of the replica finger. To illustrate the role that unidirectional versus bidirectional travel can play in the deformation mode of micropillars, images (c) (unidirectional) and (d) (bidirectional) are taken from the same sample.

Spacing also plays a large role in determining what type of failure occurs when tested arrays do not meet our survivability criteria; namely, the underlying cause of failure is always adhesion, either between pillars (lateral collapse) or between the pillars and substrate (ground collapse). The non-surviving micropillar arrays plotted throughout Fig 3 illustrate how the pairwise variety of lateral collapse generally occurs as simply the predecessor of the more drastic case of lateral collapse, which we called clustering. Pairwise lateral collapse was seen more often in the tightly-spaced arrays (and more often in the stiffer material) because micropillars that are just slender enough for a slight deflection are able to come into contact with their close neighbors and adhere to them, largely stabilizing both structures from further deflection. In contrast, the matting condition occurred only with the more flexible polymer at higher aspect ratios and the largest interpillar spacing. Such micropillars of comparatively low bending stiffness were then capable of being bent horizontally before coming into contact with their neighbors.

The form of lateral collapse known as clustering generally occurred as a more severe case of lateral collapse than the pairwise form, meaning we observed it where greater micropillar deflection could be expected. Unsurprisingly, we found that usually the bidirectional travel of the replica finger facilitated this multidirectional bending behavior. We categorized any clumping-together of greater than two micropillars as clustering but often it was seen as a distinct star-shaped cluster of seven that results from the hexagonal spacing of the arrays (Fig 4A). This example of a clustering array has the same geometric parameters (6:1 aspect ratio,

d = 6 μm, s = 2d = 12 μm) as that of the array in Fig 4B showing pairwise lateral collapse. We chose this comparison as a visual example of how, all else equal, the difference in elastic modulus between the two formulations of thiol-ene resin causes a subtle difference in the mechanical response of micropillars but markedly different outcomes as an array.

In some cases, we observed significant differences in the mechanical response of micropillars at different areas of the same test sample due to the travel path of the replica finger. For instance, Fig 4C and 4D illustrates a micropillar array sample composed of the higher modulus epoxy with 6:1 aspect ratio elements with d = 18 μm and spacing s = 2d = 36 μm. The matting that is shown in Fig 4C occurred in the first array slightly forward from where the replica finger initially makes contact. The pairwise lateral collapse shown in Fig 4D, however, took place at the pivot point in which the replica finger reached the end of its forward travel, paused, and then moved in the reverse direction. This contrast in separate locations of the same test sample is therefore between unidirectional and bidirectional travel, that is, between micropillars that the replica finger has traveled over in one direction vs. those that have been traveled over in both directions. Careful scrutiny of these images, however, reveal pairwise pillars present in Fig 4C and likewise, some pillars we considered matting in Fig 4D. This discrepancy underscores the fact that our categorization of mechanical behavior was subjective. We took great care to determine the dominant failure mode for each micropillar array tested and in most cases the particular type of adhesion was obvious. It was more difficult, however, to sort the relative severity or extent of deformation and such decisions were often a judgment call made after comparing images with those of other test samples exhibiting similar behavior.

Another example of mixed forms of mechanical behavior in the same test sample is evident in Fig 4C and 4D, where micropillars that suffered either lateral or ground collapse were surrounded by pristine micropillars. This phenomenon was extremely common; in nearly all micropillar arrays that we considered non-surviving, pristine micropillars were found throughout the replica finger's path of travel. These apparently untouched micropillars were the first indication that, rather than the bulk of our replica fingerprint ridges colliding with these relief structures, contact between the finger and the micropillar arrays largely occurred between surface asperities on the surface of the molded fingerprint ridges.

## Contact between finger mold and micropillar arrays

Examination of both the paraffin molds, taken from the index fingers of the authors in this study, and the PDMS replica fingers showed an additional length scale of microroughness smaller than that of periodic ridges. We observed these bumps and depressions regardless of the various molding, inking or impression techniques used, and while their distribution appeared to be random, they were noticeably more prevalent as asperities found on ridges (Fig 5). In light of this morphological disorder, we anticipated that our attempt at mimicking real-life interrogatory touch might complicate idealized scenarios where Hertzian or JKR contact mechanics have otherwise been used to good effect. These theories of contact can be used to predict the contact area and indentation of a spherical body into a flat counterbody provided the applied force, radius of indenter and elastic modulus of the bodies are known. In our case, we molded index finger pads that were ellipsoidal and measured ca. 20 mm × 14 mm for an area of ca. 220 $mm^2$. During testing, however, we were only able to obtain a contact area ca. 13 mm × 8 mm for an apparent contact area of ≈ 80 mm$^2$. This contact area was reached quickly during mass loading and subsequent attempts to increase this area by adding more mass only pushed this existing contact area into the substrate with more force, deflecting the substrate carrier. We noted that the width of our contact area closely matched the width the resin-printed distal phalanx that served as a "replica finger bone" and allowed us to mount the

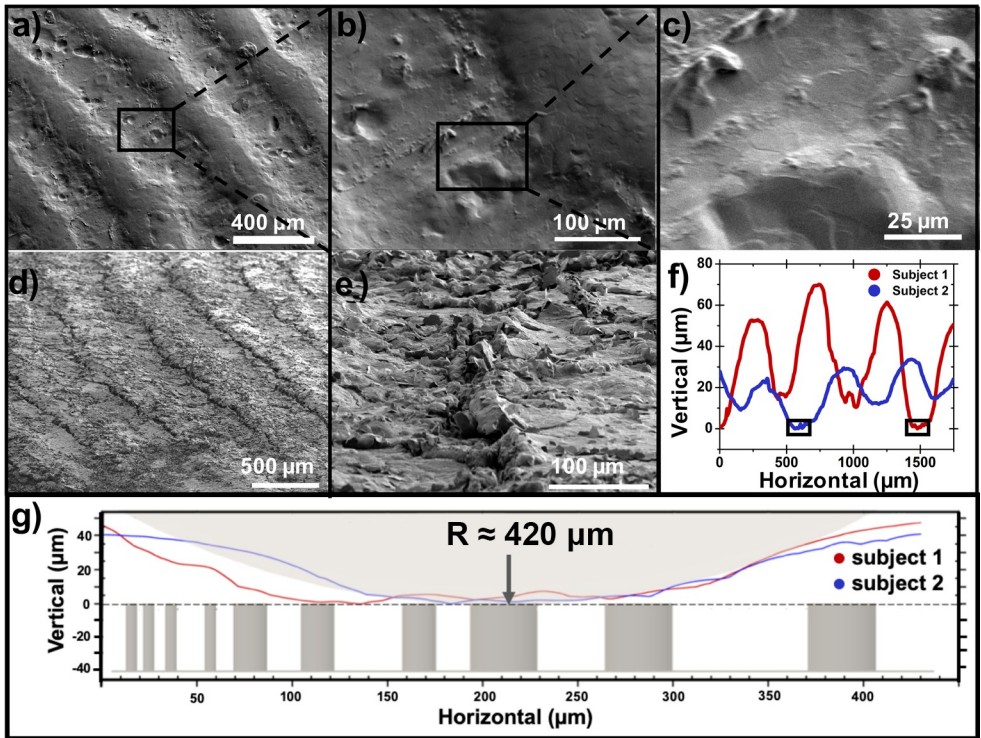

**Fig 5. Topology of human finger pads.** Panels (a) through (c) are SEM images of a paraffin mold used to cast the PDMS replica finger probes used in this study, shown at sequentially higher magnification. Note that because this mold is a negative image from which the replica finger probe is cast, pits and depressions within the lower regions represent asperities that protrude from the fingerprint ridges of the probe. Panels (d) and (e) are SEM images of a 30:1 PDMS probe at the conclusion of testing. Panel (f) shows profilometry traces of fingerprint molds from two authors to demonstrate the general dimension and variability of human fingerprints where the single ridge rectangular insets are then plotted in (g) with equally scaled axes to convey the true extent of microroughness in contact with the micropillar arrays.

replica finger. While this limitation underscores the compliance of our testing apparatus, it also reinforces how the presence of a rigid backing material such as fingernails can greatly affect the apparent contact area [55].

A similar effect has also been noted by Tomlinson et al., in a study that used real human fingers to measure friction in a variety of materials with milled surfaces [56]. They reported that after applied normal loads of 1 N, the finger seemingly met its limit of compressibility and began to behave as a rigid body. Also of note in that study is that the standard model of Hertzian contact did not fit their experimental data. Among other reasons, the authors speculated that modeling friction in fingers using a sphere in contact with a flat plate does not account for fingerprint ridges–a key feature in our own experiment. As expected, the Hertzian contact model also provided a poor fit to the contact radius and indentation depth that we measured experimentally. Due to the large difference in elastic modulus between 30:1 Sylgard-184 and both of the polymers we studied, we calculated Hertzian contact using reduced elastic modulus, plane-strain modulus, and PDMS modulus alone for a range of 100 kPa – 245 kPa found in the literature [57, 58]. The resulting Hertzian contact radius never exceeded less than half of our measured value and indentation depth was even less accurate.

On the other hand, we found agreement between our apparent contact area and the JKR contact model, which takes adhesion into account. Using representative values of surface energy reported elsewhere for UVO-treated PDMS ($\gamma_{\text{PDMS}}$ = 72 mJ $m^2$) [50] and 1%

fluorinated thiol-ene ($\gamma_{R-SH}$ = 20.2 mJ $m^2$) [59] provided values of JKR contact radius and indention depth within 1 μm of our own averaged values for some geometries. We took this as an encouraging sign that our distribution of normal force in these calculations was acceptable, especially given the assumptions we will elaborate upon shortly.

We suspect that contact in our study usually occurred between the micropillar arrays and the curved surfaces at the apex of the fingerprint ridges. These curved ridges are themselves rife with microstructures in the form of asperities that are of similar dimension to micropillars, suggesting asperity-on-asperity contact. Our experimental conditions were then best described as contact between a rough curved surface and a rough flat surface that took place at relatively high force regimes. Fig 5F shows the results of a high-resolution (24 nm per point horizontal travel) stylus profilometry scan over a horizontal range of 1.75 mm in which several consecutive ridges are shown in the approximate center of the replica finger contact area. Although the amplitude of fingerprint ranges varied considerably among co-authors, there was considerable agreement in pitch (ca. 500 μm). Curve-fitting these profilometry scans with equally scaled axes (Fig 5G) allowed us to determine an average radius of curvature for fingerprint ridges of 416 μm. We chose to include micropillars of each diameter and spacing across the x-axis to give a sense of scale between the features of the two contacting surfaces. Note that, although the profilometry traces shown speak to the roughness of fingerprint ridges, the 25 μm stylus tip is of the same order as the asperities evident in the SEM images. This means that even if the profilometry scan were to intersect with one of these random asperities, their profile would be smoothed considerably. We encourage readers who are interested in finger print ridge topology to look at the work of Childs for a 3D profilometry scan that more clearly illustrates the extent of random roughness of human finger pads [60].

In order to measure the contact width of a fingerprint ridge under load, we used fingerprint ink to ink three replica fingers molded from three different subjects and loaded them with the same 200 g mass that we used in the experiment. We made these impressions on both blank glass microscope slides as well as slides containing micropillar arrays and measured the resulting ridge widths with an optical microscope. This allowed us to determine an average fingerprint ridge contact radius, $r$ = 107.5 μm (n = 12, σ = 30.4 μm).

To further construct our contact model, we examined several representative replica fingers to determine the best way to model the ridge pattern in order to measure the total length of fingerprint ridges within the apparent contact area. We used computer-aided drafting software to construct a 3D solid model of the simplest representative pattern, featuring a slotted central "whirl" at one end of the contact area with concentric rings radiating outward according to our measured pitch (500 μm) and width ($2r$ = 215 μm). From this model we summed and averaged the inner and outer lengths of these 10 fingerprint ridges, along with the central whirl, to arrive at a total length of 115.5 mm along the central peaks of these ridges. The product of their total length and their width gave us a total area available for contact with the micropillar arrays of $215 \times 10^{-5}$ $m^2$ amounting to 26% of the apparent contact area.

In order to find the number of micropillars, $N_m$, in contact with the fingerprint ridges, we first calculated the density, n, of micropillars per unit area according to their diameter and interpillar spacing. We modeled the arrays as hexagonal unit cells with central micropillars so that their density per unit area was simply the inverse of their area, $A_{hex}$:

$$\frac{1}{n} = A_{hex} = \frac{\sqrt{3}}{2}(d+s)^2 \; m^2 \tag{4}$$

where d is the micropillar diameter and s is their spacing (we found it programmatically convenient to reformulate this simple formula in terms of our feature geometry). When we assume

top contact, the number of micropillars, $N_m$, in contact with the total area of the fingerprint ridges, $A_{ridges}$, is then:

$$N_m = n \times A_{ridges} \tag{5}$$

For instance, the micropillar arrays shown in Fig 4A and 4B with diameter $d$ = 6 μm and spacing $s = 2d$ = 12 μm have a density n of $3.56 \times 10^9$ m$^{-2}$ which gives $N_m = 8.85 \times 10^5$ microcontacts. On the other hand, an array of micropillars with diameter $d = 18\ \mu m$ and spacing $s = d = 18\ \mu m$ has a density $n = 8.9 \times 10^8\ m^{-2}$ giving $N_m = 2.21 \times 10^4$ microcontacts. We borrowed a straightforward approach that has been applied elsewhere [47] and took these contacts as a number of points upon which to distribute the total applied normal force in our contact model. The agreement of our contact geometry with that predicted by the JKR model thus leads us to believe that this normal force distribution was reasonably accurate. Although both approaches are generally applied to stationary contact, the total population of asperity junctions is conserved in a multicontact interface under conditions of steady sliding and constant normal force [61].

The distribution of tangential force that provided transverse (shear) loading on individual micropillars was a matter of greater difficulty. We assumed that contact with micropillars would occur only with the leading edge of fingerprint ridges and as the replica finger traveled, the contact population would be constantly refreshed as with contacts bearing axial stress. We therefore multiplied the linear length of the modeled fingerprint ridges within our apparent contact area by the linear density of micropillars to arrive at a micropillar population in contact with those ridges. Illustrated details of our contact model geometry can be found in S2 Fig.

This manner of modeling a force distribution is a decidedly coarse approach to a matter of considerable complexity, as tangential force and motion in cases such as this generally require examination of friction in a multicontact interface. When such an interface involves a viscoelastic material (such as PDMS or skin), there are additional aspects to be considered such as junction rheology and the dependence of frictional response on velocity of travel and geometric age. While we have attempted to better understand the bending deflection of micropillars in what follows, rubber friction and the contact mechanics of rough on rough surfaces are outside the scope of this study. This is largely because the necessary material properties–e.g. complex modulus, E(ω), of 30:1 PDMS, surface roughness power spectrum, C(q), of the combined elastic half-space–were not available [62]. We, however, wish to encourage interested readers to look at a comprehensive review on the topic of solid friction [61].

## Details of contact and deformation

We were able to measure the total tangential force applied by the linear actuator using a force sensor positioned at the fixed end of the device and opposing the moving stage so that its applied force ($F_T \approx$ 95 N) is measured when its forward movement is completely arrested. We likewise have high confidence that the total applied normal force ($F_N$ = 2.08 N) was the 200 g of slotted calibration mass in addition to the mass of the replica finger itself ($\bar{m}_{finger} = 12.4g, n = 4$). The manner in which these total forces were distributed to individual micropillars, however, was uncertain and cannot be assumed purely on a geometric basis due to the randomly rough nature of human finger pads. But due to our inability to observe the interface of replica finger and micropatterned features *in situ*, our quasi-static analysis was based on the simple geometric contact model detailed in the previous section. This quasi-static analysis was, however, explicitly statically indeterminant because we were unable to solve for an equilibrium of moments and forces, i.e., the shape of the elastic line was known but the moments and forces were not [63]. Ultimately, our understanding of the mechanics at the

scale of individual micropillars was necessarily informed by the deformation observed *ex situ* through SEM image analysis of micropillar deformation. Because we are able to observe the slope and deflection of micropillars subjected to bending stress, we were able to solve for the inverse problem of likely distributions of normal and tangential force that caused this deformation.

When treating a micropillar as a column with one fixed end and the other end free, it can be modeled as a cantilever beam. When such a beam is subjected to both an axial load and a transverse load simultaneously, it is considered a beam-column [64] and the standard forms of either Timoshenko or the Euler-Bernoulli equations for beam bending do not apply. This is because the axial compression can greatly increase the bending moment at the base of the beam, which in turn affects its slope and deflection. In the case of the beam-column, the deflection, y, of deformed micropillars is given by [65–67]:

$$
y = \frac{-W}{kP} \left[ \frac{sin\ kl[1 - cos\ k(l - a)] - cos\ kl[k(l - a) - sin\ k(l - a)]}{cos\ kl} \right] +
$$
$$
\frac{W}{kP} \frac{1 - cos\ k(l - a)}{cos\ kl} sin\ kx - \frac{W}{kP}[k\langle x - a \rangle - sin\ k\langle x - a \rangle]
$$

(6)

where W is the applied concentrated transverse (tangential) load, P is the applied axial (normal) load, a is the distance from the tip that W is applied, *l* is the length of the beam-column, and $k = \sqrt{P/EI}$ is a parameter used to factor in the effect of applied normal force and bending stiffness. The expression $<x - a>$ is a unit step function that that is designed such that:

$$
\langle x - a \rangle^n = 0\ if\ x < a
$$
$$
\langle x - a \rangle^n = \langle x - a \rangle^n\ if\ x > a
$$
$$
\langle x - a \rangle^n = undefined\ when\ x = a
$$

(7)

Although we believe that a distributed load describes more accurately the manner in which tangential force is delivered to the micropillars, the concentrated transverse force equation is more compact. A distributed load can be equivalently expressed as a concentrated load where the integrated area of the distributed force profile is applied at the centroid of the shape of that profile. In choosing to use Eq (6), we were then able to easily experiment with different force profiles by simply varying the *a* parameter. Fig 6A shows a free-body diagram of our beam column model with relevant mechanical parameters and the boundary parameters that are a result of the end constraints.

We were thus able to conduct analysis of SEM images to determine the deflection, and constructed 3D models of the deformed micropillars so we could view them from various angles (including the 30˚ tilt angle we used for all SEM images) and refine the accuracy of our observations to account for parallax error. Fig 6B shows 3D models of the micropillars with $d = 6$ $\mu m$ and $s = 2d = 12\ \mu m$, SEM images of which were shown in Fig 4A and 4B, along with the normal and axial forces that were required to cause the given deflections.

In the case of the more compliant polymer system ($E = 10\ MPa$), we found that all of the aspect ratios that we tested experienced a normal force that exceeded the critical force, $P_{cr}$, for elastic instability [63, 68], namely:

$$
P_{cr} = \frac{\pi^2 EI}{4l^2}
$$

(8)

When such a structure is subjected to an axial load $P > P_{cr}$, not only will it fail to return elastically to its original undeformed shape when the external force is removed, but it will be in

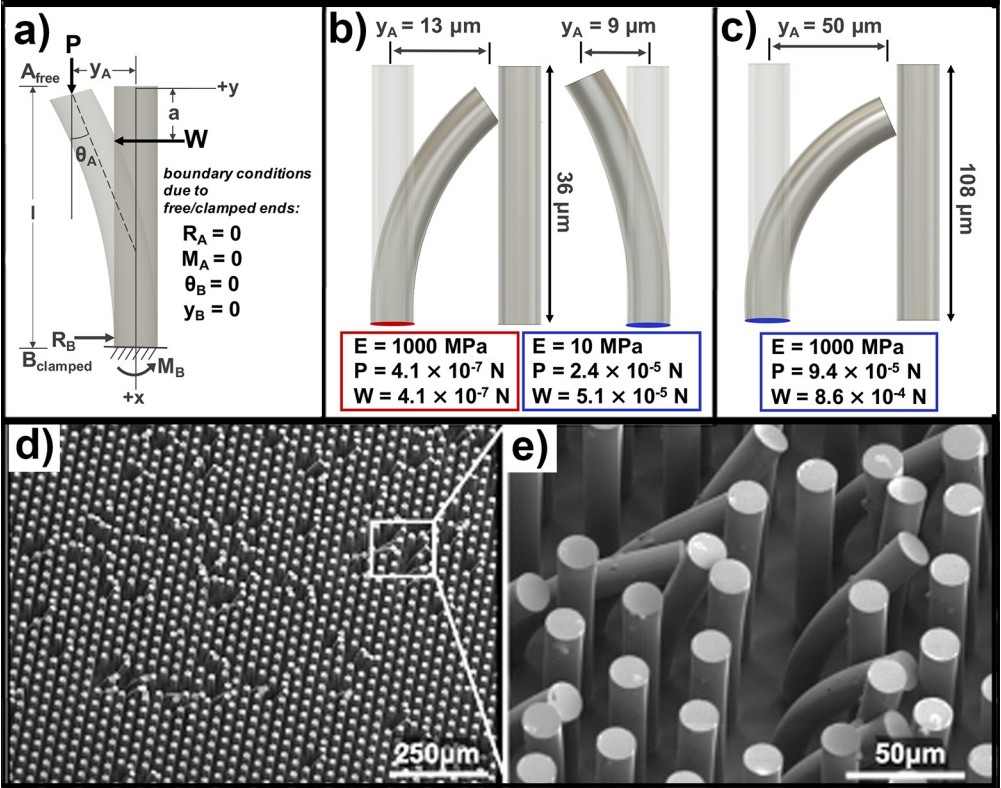

**Fig 6. Mechanical analysis of PDMS-micropillar interactions.** (a) Free-body diagram with associated moments, reactions, forces and deformation. (b) 3D models of the micropillar cases from Fig 4A and 4B along with deflections and forces to produce those deflections. (c) 3D model showing same for the $d$ = 18 $\mu m$, $s$ = $d$ = 18, $\mu m$, $E$ = 1000 $MPa$ case. (d-e) SEM images showing the isolated nature of deformation for the case shown in (c).

unstable equilibrium. This means that the equations based on static equilibrium such as (6) are not strictly applicable because equilibrium might be disrupted by an infinitesimal bending of columns subjected to compression from P > $P_{cr}$. So not only will such micropillars be prone to large deflection (as already indicated by their much lower elastic modulus), but they will be especially vulnerable to the adhesion phenomena we observed (i.e. clustering, in this particular case).

We therefore posit that our single micropillar mechanical model predicts a much lower force to deflect these lower modulus micropillars simply because their spacing prevents them from bending further. The two micropillar cases shown in Fig 4A and 4B with identical geo-metric parameters but different elastic moduli were presumed to experience the same magni-tude of forces because the images were taken from the same region along the replica finger path of travel. But substituting the forces necessary to deflect the E = 1000 MPa micropillars a distance $y_a$ = 9 $um$ into (6) at $E$ = 10 $MPa$ yields a non-physical amount of deflection that we would interpret as ground collapse. But perhaps because they were bent into contact with their neighboring elements, these more compliant micropillars adhere together into a structural configuration that was much stronger and therefore proof against further deformation.

We ultimately found that micropillars composed of the more compliant polymer were pro-hibitively sensitive to the manner of combining axial and transverse loading used in (6). We generally used the axial load calculated from our simple contact model because this approach had precedent and agreed well with JKR contact mechanics. When we tried to solve for the

transverse loading that would lead to the observed deflection profile, the elastically unstable beam-column would either deflect negligibly or explosively. We suspect this is because the assumption of axial-centric loading in (6) leads to buckling modes without deflection at the free end, reminiscent of classical Euler buckling with maximum deflection at the midpoint of a column.

For the sake of arriving at a solution for combined forces that could have resulted in an end-point deflection of 13 $\mu m$, we instead modeled the $E = 10$ $MPa$ micropillar shown in Fig 6B as if it were obliquely and eccentrically loaded [69]. We chose a resultant force, P, that was directed at an angle $\alpha = 45° = \pi/4$ from vertical and centered at a point with eccentricity $e = 0.5r = 3$ $\mu m$ radially outward from the centroid of the cross-section. After applying the same boundary conditions as in (6) for a cantilevered beam-column, the deflection has the solution

$$y(x) = [\cot(\alpha)/k]\sin(kx) + [(y_A + e) - l\cot(\alpha)]\cos(kx) + [\cot(\alpha)(l - x) - (y_A + e)] \quad (9)$$

where $y_a$ is the deflection at the free end and $l$ is the length of the beam column (as before). In this case, however, the parameter $k = \sqrt{P\sin \alpha/EI}$ so that only the axial compressive component of the oblique eccentric load is included. This eccentricity adds an additional bending moment while the oblique loading angle combines the axial and transverse loads at the expense of applying the transverse load to the side of the structure. The resulting elastic line, however, showed a smooth bending profile equivalent to what we observed under SEM.

## Guidelines for survivability

Our experiment was motivated by the hypothesis that there would be a window of diameter, spacing, aspect ratio and elastic modulus that would allow an array of microfabricated structures to survive being touched firmly by people and this window could provide useful guidelines for the rational design of such surfaces. The column plot shown in Fig 7 illustrate the sum of mechanical outcomes for the parameters we tested. We chose to assign the color yellow to the occurrence of ploughing because, although we consider it survival of the micropillar array, it is reasonable to assume there may be applications where the possibility of ploughing is undesirable.

The survivability results shown in Fig 7 highlight several trends. The more severe failure modes were seen in the more compliant (E = 10 MPa) system while the stiffer (E = 1000 MPa) material shows a prevalence of pairwise lateral collapse in pink. The larger population of yellow segments in the stiffer polymer plot illustrates the tendency of those micropillars to resist deformation by ploughing material off of the replica finger while (we suspect) many of the more compliant micropillars of same geometry avoid ploughing owing to elastic deformation and recovery. There is also more variability in the column plot representing micropillars with E = 10 MPa; this variability reflects the elastic instability of micropillars at the lower modulus. In each plot, the saturation in the red palette increases from top-right to bottom-left showing the trend towards failure as diameter decreases and spacing increases.

We intended this study to inform the rational design of micropatterned surfaces that could be handled by human users. Most design decisions involve constraints that recommend a particular attribute in order to grant the best chance of survivability. For example, if an engineering application called for 6:1 micropillars that have a smaller diameter but significant interpillar spacing (e.g. $d = 6$ $\mu m$ $and$ $s \leq 2d = 12$ $\mu m$), a higher modulus polymer with $E > 1000 MPa$ would be in order for increased durability and resistance to lateral collapse when handled. If design constraints called for a lower aspect ratio and spacing, but required a larger diameter (e.g. $d = 18$ $\mu m$), then a lower modulus polymer would be the more appropriate choice if the potential for ploughing is to be avoided.

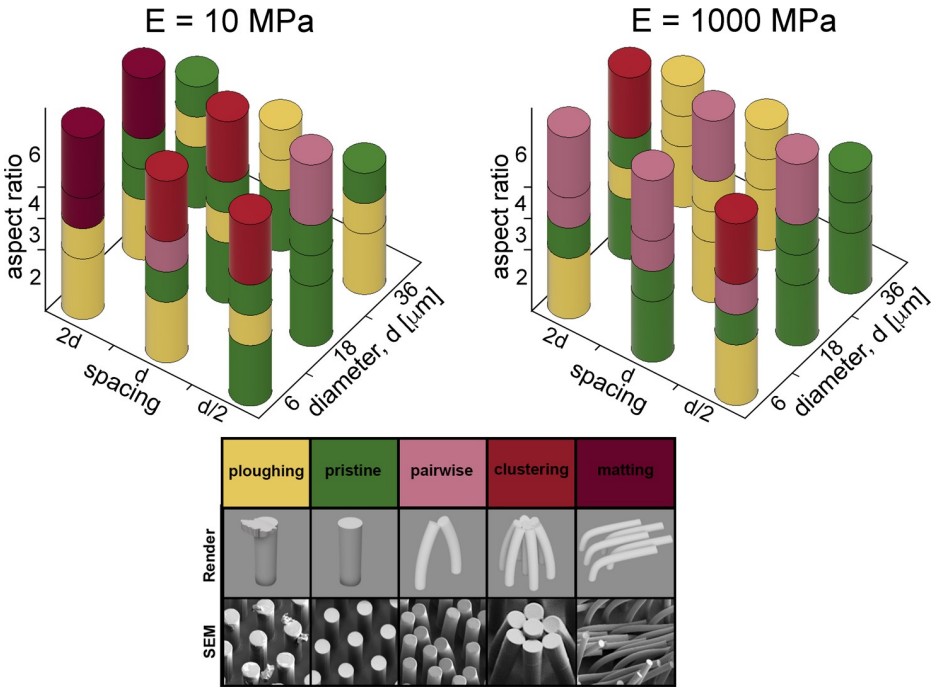

**Fig 7. Resulting mechanical outcomes for all samples tested.** Mechanical outcomes for all E = 10 MPa micropillar samples are shown on the left while all E = 1000 MPa micropillar results are shown on the right. The legend in the foreground assigns reds of increasing saturation for increasingly severe adhesion/deformation. Green indicates the tested micropillars remained in pristine condition while yellow was chosen for ploughing because micropillars survived despite injurious results for the replica finger.

All else equal, lower aspect ratio structures and larger diameters lead to increased survivability. Tighter spacing means a higher micropillars density. Decreased pitch increases the real contact area of the multicontact interface resulting in a decreased local pressure that each micropillar must bear. The tradeoff with tighter spacing is that micropillars then have less room for deflection without coming in contact with, and adhering to, neighboring elements.

Because polymers—particularly adhesive resins of relatively high surface energy—were the material of interest in our study, the failure modes were all related to bending deflections leading to interpillar adhesion. Under a given loading condition, the stiffness, K, of an individual micropillar,

$$K = \frac{P}{y} = E\frac{A}{l} \tag{10}$$

determines whether that micropillar is capable of bending deflection to a particular distance for interpillar adhesion. This distance is largely dependent on the spacing between the individual elements. In the event that this distance is greater than the length of the micropillar, under a sufficiently high load, that micropillar will continue bending until adhering to the substrate by ground collapse. In any case, with this simple dependency of polymer micropillar failure upon stiffness and spacing, s, optimization should be possible, for example by minimizing the elastic modulus subject to the constraint that the bending deflection,

$$y < y_{critical} = s \tag{11}$$

This relation is admittedly a first-order approximation, and depends on a reasonably accurate distribution of forces in the system. A more sophisticated formulation was developed by Glassmaker et al. by balancing energies of elastic deformation and adhesion [32]. Glassmaker also offered insight into predicting the transition from doublet to clump formation (which we refer to as pairwise and clustering lateral collapse, respectively) and even successfully predicted the characteristic clump size for their array and element geometry. We must note that their experimental validation consisted of spherical indentation with a smooth glass probe ($r = 4$ $mm$) at speeds of both 1 and 10 μm s$^{-1}$ so it did not feature lateral movement or human-scale speeds. This type of testing, however, allowed them to make appropriate use of JKR contact mechanics in their model and avoid many complications inherent in the sliding of rough on rough surfaces. We therefore believe that the JKR contact model has limited applicability in our own experiment for reasons which we have already detailed as well as the observation that deformation we observed was not particularly confined to the center of the contact radius (where JKR dictates pressure is maximum) as it was in their scenario.

## Conclusions

Experiments in contact mechanics generally take place in idealized conditions using lower than human-scale forces that are often necessary to reveal subtle adhesion phenomena. Studies in friction largely take place at lower than human-scale velocities to avoid temperature effects. Both disciplines tend to constrain their experiments in this way to increase precision and limit sources of error. Our experiment illustrates that a motion as simple as sliding a finger across an engineered textured surface can be complicated to understand quantitatively, especially if it cannot be observed in situ and with fully characterized materials and an expensive sophisticated test apparatus. Our intent was to mimic interrogatory touch in sliding forward, pausing and sliding backways across a micropattened surface in a manner that is inconvenient for isolating physical phenomena but represents a realistic occurrence if such surfaces engage the curiosity of primates.

We have determined that for the two polymers we chose study, failure occurred entirely due to adhesion via lateral collapse or ground collapse. For two of the diameters we tested, $d = 6$ $µm$ and $d = 18$ $µm$, this failure took place at an aspect ratio of 6:1 due to the gradual decrease in bending stiffness as length increases. At equal aspect ratios, a larger cross-sectional area grants an increased chance of survivability. In light of this fact, we therefore suspect that survivability at the 6:1 aspect ratio may begin at or around $d = 36$ $µm$ due to the absence of deformation for those diameter arrays at 4:1. Unfortunately, we were unable to realize fabrication of those arrays so this prediction cannot be confirmed.

Because the failure mechanisms are driven by competition between elastic restoring force and adhesion/surface energy, the question of whether adhesion can be mitigated would be an enriching topic of future work. In haptics, the potential to make adhesion and thus, changes in texture arising from lateral/ground collapse, reversible would open up a wide range of applications and commercial potential.

When the replica finger we used in this study was stationary, as during loading prior to movement and during the one second pause between sliding forward and in reverse, understanding was largely a problem of viscoelastic contact mechanics between rough on rough surfaces. While the finger was in motion, the scenario was one of rubber friction in a system that was likely overdamped by finite device stiffness. We have posited a model employing the mechanics of cantilevered beam-columns in order to predict the mechanical response of micropillars being deflected towards failure by adhesion. Future work is needed to quantitatively understand the interaction of human fingers with micropatterned surfaces, and great

care should be taken to select materials that are highly characterized, especially the viscoelastic material chosen to mimic skin. Human fingertip mechanical properties are also highly influenced by skin hydration [70] and while this would require further environmental controls, it would be insightful to investigate the effects of lubrication at the interface.

## Supporting information

**S1 Fig. Experimental apparatus setup.** (a) Solid model of test apparatus with annotation of key features. Inset shows replica finger on drive shaft with tangential and normal force sensors. (b) Photograph of test apparatus with inset showing approximate position of replica finger prior to contact with substrate and mass loading.
(TIF)

**S2 Fig. Contact model geometry.** (a) Our determination of indentation depth from measured quantities for average fingerprint ridge radius and average fingerprint ridge width. (b) Solid modeling using Autodesk Fusion 360 allowed us to sum the lengths of fingerprint ridges and to solve for total fingerprint ridge area for the simple representative geometry shown. (c) Micropillar density is the inverse of the area of a hexagonal unit cell, a function of diameter and spacing. We were then able to solve for the total number of microcontacts as product of total fingerprint ridge area and micropillar density.
(TIF)

**S3 Fig. Location of transverse load, W.** Transverse load W is applied across span of length a where a is determined by the lessor value of spacing or indentation depth. The inset showing forces applied to a deflecting micropillar are simplified for clarity. In our calculations, the actual distance of a concentrated transverse load W is a/2 from the free end to approximate a distributed load with rectangular profile applied along length a.
(TIF)

**S4 Fig. Representative images from the entire series of micropillars with d = 18 μm.** Samples composed of the E = 1000 MPa polymer are shown in the blue border and those composed of the E = 10 MPa polymer are shown in red border. Aspect ratios from 2:1 to 6:1 are denoted in the upper-left corner of each image. Micropillar arrays with spacing s = d/2 = 9 μm are shown in (a). Those with spacing s = d = 18 μm are shown in (b) while those with spacing s = 2d = 36 μm are shown in (c).
(TIF)

**S1 Appendix. Micropillar array fabrication process.** Details included for each process step.
(PDF)

**S2 Appendix. Additional error analysis on contact model calculations.**
(PDF)

**S3 Appendix. Finite element analysis.**
(PDF)

## Author Contributions

**Conceptualization:** Mickey Finn, III, Charles Dhong, Darren J. Lipomi.

**Data curation:** Mickey Finn, III, Jeremy Treiber, Darren J. Lipomi.

**Formal analysis:** Mickey Finn, III, Jeremy Treiber, Darren J. Lipomi.

**Funding acquisition:** Darren J. Lipomi.

**Investigation:** Mickey Finn, III, Jeremy Treiber, Mahmoud Issa, Christian J. Martens, Colin P. Feeney, Lehna Ngwa.

**Methodology:** Mickey Finn, III, Jeremy Treiber, Charles Dhong, Darren J. Lipomi.

**Project administration:** Mickey Finn, III, Charles Dhong, Darren J. Lipomi.

**Resources:** Darren J. Lipomi.

**Software:** Mickey Finn, III, Jeremy Treiber, Charles Dhong, Darren J. Lipomi.

**Supervision:** Mickey Finn, III, Darren J. Lipomi.

**Validation:** Mickey Finn, III, Darren J. Lipomi.

**Visualization:** Mickey Finn, III, Jeremy Treiber, Darren J. Lipomi.

**Writing – original draft:** Mickey Finn, III, Jeremy Treiber, Darren J. Lipomi.

**Writing – review & editing:** Mickey Finn, III, Jeremy Treiber, Charles Dhong, Darren J. Lipomi.

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
