## [Decision Letter · Decision Letter 0]

11 Apr 2021

PONE-D-21-08649

Survival of Polymeric Microstructures Subjected to Interrogatory Touch

PLOS ONE

Dear Dr. Lipomi,

Thank you for submitting your manuscript to PLOS ONE. After careful consideration, we feel that it has merit but does not fully meet PLOS ONE’s publication criteria as it currently stands. Therefore, we invite you to submit a revised version of the manuscript that addresses the points raised during the review process.

Reviewer 1

In this article, the Authors presented the analysis of various deformation of polymeric micro pillar structures for applications in haptic systems. To describe effects of each parameter (modulus, spacing, diameter and aspect ratio) on the survival of the structures, the author employed theoretical models of various deformation phenomena (ex. Lateral collapse, Matting, Ground collapse) from other literatures. Although this work is interesting analysis about deformation of microstructures, the paper should be addressed to critical major issues to be published under PLOS ONE.

I. In this paper, the Authors present mathematical models for several damages of microscopic pillar structures, induced by applied stresses. However, these models are basically proposed in other literatures… This study does not seem to have any new theoretical advances from existing mathematical models. It seems that external haptic stimuli can be an additional variable.

II. In Figure 3, the Authors analyzed how the modulus, aspect ratio, diameter and spacing of micro-pillars affect the deformation phenomenon. However, this data seems very complex to interpret. I recommend organizing the all evidence (e.g. OM or SEM images) for the results in supporting information, according to the size and arrangement of various microstructures.

III. I don't know what the meaning of the data in Figure 4 is. Rather than listing examples of simple collapse, I recommend theoretically analyzing the correlation between the degree of damage and the degree of stress applied.

IV. The authors used the finger replica (PDMS) when conducting the experiments. However, the Authors should analyze the similarities (e.g. strength, surface energy, and fingerprints) of the replica of the finger to replace a real finger.

Reviewer 2

In this work, the authors reported the collapse of polymer microstructures under external mechanical perturbations. As the authors described, high aspect ratio polymeric microstructures have a wide range applications in diverse engineering fields. Thus, the study on their mechanical stability is of great importance. Although such studies on the stability of polymeric microstructures have been extensively studied during the heyday of soft lithography, the authors performed more advanced, in-depth theoretical and experimental analyses by considering practical situations. The overall results and analyses are reasonable. Thus, I recommend this work to be accepted without significant alterations. The one minor comment is that the manuscript is quite long. I recommend that the authors consider if the length of the manuscript  can be reduced for more conciseness of the manuscript.

We look forward to receiving your revised manuscript.

Kind regards,

Tae-il Kim

Academic Editor

PLOS ONE

Journal Requirements:

2.  Thank you for including your ethics statement:  "Collection of participant samples (fingerprints) was done under the supervision of the University of California, San Diego Human Research Protections Program Institutional Review Board under project #191950S. Informed written consent was obtained for collection of participant samples."

Please amend your current ethics statement to confirm that your named institutional review board or ethics committee specifically approved this study.

"This work was supported by the National Science Foundation under grant number CBET-

 1929748. Additional support was provided by the Center for Wearable Sensors in the Jacobs

School of Engineering at 777 the University of California San Diego, and member companies Honda,

Dexcom, Samsung, Huami, PepsiCo, Gore, Sony, Corning, and Merck KGaA. This work was

performed in part at the San Diego Nanotechnology Infrastructure (SDNI), a member of the

National Nanotechnology Coordinated Infrastructure, which is supported by the National

Science Foundation (grant ECCS-1542148)."

Additionally, because some of your funding information pertains to commercial funding, we ask you to provide an updated Competing Interests statement, declaring all sources of commercial funding.

In your Competing Interests statement, please confirm that your commercial funding does not alter your adherence to PLOS ONE Editorial policies and criteria by including the following statement: "This does not alter our adherence to PLOS ONE policies on sharing data and materials.” as detailed online in our guide for authors  http://journals.plos.org/plosone/s/competing-interests.  If this statement is not true and your adherence to PLOS policies on sharing data and materials is altered, please explain how.

Please include the updated Competing Interests Statement and Funding Statement in your cover letter. We will change the online submission form on your behalf.

5. Please include captions for *all* your Supporting Information files at the end of your manuscript, and update any in-text citations to match accordingly. Please see our Supporting Information guidelines for more information: http://journals.plos.org/plosone/s/supporting-information.

Reviewers' comments:

Reviewer's Responses to Questions

**Comments to the Author**

1. Is the manuscript technically sound, and do the data support the conclusions?

Reviewer #1: Partly

Reviewer #2: Yes

2. Has the statistical analysis been performed appropriately and rigorously? 

Reviewer #1: Yes

Reviewer #2: N/A

3. Have the authors made all data underlying the findings in their manuscript fully available?

Reviewer #1: Yes

Reviewer #2: Yes

4. Is the manuscript presented in an intelligible fashion and written in standard English?

Reviewer #1: Yes

Reviewer #2: Yes

5. Review Comments to the Author

Reviewer #1: In this article, the Authors presented the analysis of various deformation of polymeric micro pillar structures for applications in haptic systems. To describe effects of each parameter (modulus, spacing, diameter and aspect ratio) on the survival of the structures, the author employed theoretical models of various deformation phenomena (ex. Lateral collapse, Matting, Ground collapse) from other literatures. Although this work is interesting analysis about deformation of microstructures, the paper should be addressed to critical major issues to be published under PLOS ONE.

I. In this paper, the Authors present mathematical models for several damages of microscopic pillar structures, induced by applied stresses. However, these models are basically proposed in other literatures… This study does not seem to have any new theoretical advances from existing mathematical models. It seems that external haptic stimuli can be an additional variable.

II. In Figure 3, the Authors analyzed how the modulus, aspect ratio, diameter and spacing of micro-pillars affect the deformation phenomenon. However, this data seems very complex to interpret. I recommend organizing the all evidence (e.g. OM or SEM images) for the results in supporting information, according to the size and arrangement of various microstructures.

III. I don't know what the meaning of the data in Figure 4 is. Rather than listing examples of simple collapse, I recommend theoretically analyzing the correlation between the degree of damage and the degree of stress applied.

IV. The authors used the finger replica (PDMS) when conducting the experiments. However, the Authors should analyze the similarities (e.g. strength, surface energy, and fingerprints) of the replica of the finger to replace a real finger.

Reviewer #2: In this work, the authors reported the collapse of polymer microstructures under external mechanical perturbations. As the authors described, high aspect ratio polymeric microstructures have a wide range applications in diverse engineering fields. Thus, the study on their mechanical stability is of great importance. Although such studies on the stability of polymeric microstructures have been extensively studied during the heyday of soft lithography, the authors performed more advanced, in-depth theoretical and experimental analyses by considering practical situations. The overall results and analyses are reasonable. Thus, I recommend this work to be accepted without significant alterations. The one minor comment is that the manuscript is quite long. I recommend that the authors consider if the length of the manuscript can be reduced for more conciseness of the manuscript.

6. PLOS authors have the option to publish the peer review history of their article (what does this mean?). If published, this will include your full peer review and any attached files.

Reviewer #1: No

Reviewer #2: No

---

## [Author Response · Author response to Decision Letter 0]

16 Jul 2021

(the following is also appended to the submitted revision cover letter)

Reviewer 1:

Recommendation: Major revisions needed as noted.

Reviewer 1, Summary Comment – In this article, the Authors presented the analysis of various deformation of polymeric micro pillar structures for applications in haptic systems. To describe effects of each parameter (modulus, spacing, diameter and aspect ratio) on the survival of the structures, the author employed theoretical models of various deformation phenomena (ex. Lateral collapse, Matting, Ground collapse) from other literatures. Although this work is interesting analysis about deformation of microstructures, the paper should be addressed to critical major issues to be published under PLOS ONE.

Our response – We thank the reviewer for their careful review of our manuscript. We hope the changes below are satisfactory.

Reviewer 1, Comment 1 – In this paper, the Authors present mathematical models for several damages of microscopic pillar structures, induced by applied stresses. However, these models are basically proposed in other literatures. This study does not seem to have any new theoretical advances from existing mathematical models. It seems that external haptic stimuli can be an additional variable.

Our response – We thank Reviewer 1 for their insight into how the novelty of this study might be perceived by readers. We agree that employing classical relations originally developed to describe the mechanical stability and deformation of larger-scale structures, such as beam-column statics, to micro- and nano-scopic structures is common in other publications. We assume this is so because these relations have provided good agreement with experimental observations and there is no reason to believe that they do not scale to microscopic dimensions. 

For convenience, we have reproduced Figure 2 and the accompanying travel path of the replica finger, which we believe encapsulates most types of sliding motion (contact, forward motion, stopping, and backward/retreading).

(unable to include Fig 2 in online Response to Reviewers; please see Revision Cover Letter)

 Figure 2. Summary of the experimental design. (a) Surface of PDMS replica finger with fingerprint ridges molded from human fingers. (b) Overview of geometrical parameters, namely micropillar diameter, aspect ratio and interpillar spacing, that were varied in this experiment. (c) Rendering of replica finger showing the sliding path across substrate carrier. The inset shows example SEM images of the phenomena we refer to as pair-wise, clustering, and matting. A complete schematic drawing of the testing apparatus can be found in the Supplementary Information. 

We agree that adding additional external haptic stimuli would be quite interesting and would make a worthwhile endeavor for future work. We feel that adding additional external haptic stimuli to this study—beyond the forward and reverse sliding motion of the replica finger—has a high chance of confounding variables. Varying three different geometric parameters as well as one material property already complicates the role that each variable plays in survivability. We chose to include the reverse travel in our test motion because, while it complicates our analysis, it exemplifies the natural way in which a person might interact with a textured surface while handling a haptic display or an object with a specialized coating. As our study was primarily focused on such a realistic occurrence, we considered this test motion to be worthwhile to examine. As interesting as it would be to include any additional factors, such as additional modes of external haptic stimuli, we felt that our manuscript was already quite long (as noted by Reviewer 2), and decided to limit our analysis to sliding/retreading as described in Figure 2. Nevertheless, we agree with Reviewer 1 that it is a good idea to invite the reader to consider additional modes of haptic contact, perhaps as the topic of future work.

Our change to the MS (Conclusions, section starting at line 744) – 

From: Our intent was to mimic interrogatory touch in sliding forward, pausing and sliding backways across a micropattened surface in a manner that is inconvenient for isolating physical phenomena but represents a realistic occurrence if such surfaces engage the curiosity of primates.

To: Our intent was to mimic interrogatory touch in sliding forward, pausing and sliding backways across a micropattened surface in a manner that is inconvenient for isolating physical phenomena but represents a realistic occurrence if such surfaces engage the curiosity of primates. These modes of interaction were intended to capture as much of the complexity of interacting with a device (e.g., a smartphone) as possible, without introducing additional variables that may have added substantial complexity to the analysis. In the future, it would be worthwhile to investigate additional modalities of haptic interaction, for example, variable pressure, tapping, swirling, and scratching.

Reviewer 1, comment 2 – In Figure 3, the Authors analyzed how the modulus, aspect ratio, diameter and spacing of micro-pillars affect the deformation phenomenon. However, this data seems very complex to interpret. I recommend organizing the all evidence (e.g. OM or SEM images) for the results in supporting information, according to the size and arrangement of various microstructures.

Our response – We thank Reviewer 1 for their perspective, as we struggled with how to present the results with four independent variables. Ultimately, we constructed two different plots to summarize our findings: Figure 3 with the most dominant independent variable, separated into multipanels according to another important variable (diameter), and the spectrum of mechanical outcomes as the dependent variable was meant as a more fine-grained synopsis of our results. Figure 4, on the other hand, was meant to be courser and provide an “at a glance” summary of our findings. These figures are reproduced below for convenience.

(unable to include Fig 3 in online Response to Reviewers; please see Revision Cover Letter)

Figure 3. Survivability of a selection of tested micropillar arrays. Panel (a) displays the smallest diameter micropillars (d = 6 �m) that were the most prone to deformation and the only diameter to display lateral and ground collapse phenomena at both 4:1 and 6:1 aspect ratios. In contrast, panel (b) shows the outcome for micropillar arrays with the largest diameter (d = 36 �m) for both materials. The medium diameter (d = 18 �m) micropillars for the more compliant material (E = 10 MPa) are shown in (c) while the medium diameter micropillars composed of the stiffer polymer (E = 10 MPa) are shown in subpanel (d).

Figure 4. Representative SEM images showing the deformation of micropillar arrays. Panel (a) shows an example of “clustering” lateral collapse by the 6 �m diameter elements for the compliant (E=10 MPa) material. In contrast, panel (b) shows the stiffer material (E=1000 MPa) with the same geometry exhibiting pairwise lateral collapse. Both images (a) and (b) were taken in the bidirectional region of the travel path of the replica finger. To illustrate the role that unidirectional versus bidirectional travel can play in the deformation mode of micropillars, images (c) (unidirectional) and (d) (bidirectional) are taken from the same sample.

Granted, these figures provide only a summary (Figure 3) and snapshot (Figure 4) of the range of mechanical behavior observed. However, according to PLOS ONE policy, all of our raw data will be freely available. Presenting all of our data, by way of figures in the supporting data, would be tremendously difficult (especially when the data will already be available elsewhere). The reason for this is that the conclusions we have reached are, by no means, compact nor less difficult to parse than the results we have offered in the text and figures. The reason for this is that there are a multitude of SEM images and the conclusions we reached often involved double- and triple-checking these images of each sample against others having similar parameters or mechanical outcomes. As we began collecting this data and realizing the challenge of placing datapoints not just according to general mechanical outcome, but also the severity of those outcomes, we began taking progressively more images. The results we have presented in this study are therefore the result of many ~1000 SEM images and would not be suitable to include in the SI in its entirety. In the interest of brevity, we admitted that our placement of each datapoint required a judgement call (lines 419-428), reproduced here for convenience. 

“This contrast in separate locations of the same test sample is therefore between unidirectional and bidirectional travel, that is, between micropillars that the replica finger has traveled over in one direction vs. those that have been traveled over in both directions. Careful scrutiny of these images, however, reveal pairwise pillars present in Fig 4c and likewise, some pillars we considered matting in Fig 4d. This discrepancy underscores the fact that our categorization of mechanical behavior was subjective. We took great care to determine the dominant failure mode for each micropillar array tested and in most cases the particular type of adhesion was obvious. It was more difficult, however, to sort the relative severity or extent of deformation and such decisions were often a judgment call made after comparing images with those of other test samples exhibiting similar behavior.”

In the interest of brevity, we spared the reader a recounting of how each subsequent datapoint resulted in a reexamination, and oftentimes, a slight adjustment of the relative positions of the surrounding datapoints so that they would all be visible while being correctly ordered. Nevertheless, we wholeheartedly agree with the reviewer that a somewhat more complete figure containing a subset of the raw images should be added. This figure will bridge the data reduction performed in the main text to the raw data that will be available in accordance with the policies of PLOS ONE. 

Our change to the SI – We have added a multipanel figure in the Supplementary Information illustrating all the samples of the middle diameter (d = 18 µm). This set includes a wide range of failure modes as well as samples that met our survivability criteria and should provide evidence of our conclusions while being compact enough not to overwhelm readers with a multitude of magnifications and locations. 

(unable to include S7 Figure in online Response to Reviewers; please see Revision Cover Letter)

S7 Figure. Representative images from the entire series of micropillars with d = 18μm. Samples composed of the 1000 MPa polymer are shown in blue border and those composed of the 10 MPa polymer are shown in red border. Aspect ratios from 2:1 to 6:1 are shown in the upper-left corner of each image. Micropillar arrays with spacing s = d/2 = 9 μm are shown in (a). Those with spacing s = d = 18 μm are shown in (b) while those with spacing s = 2d = 36 μm are shown in (c).

Reviewer 1, comment 3 – I don't know what the meaning of the data in Figure 4 is. Rather than listing examples of simple collapse, I recommend theoretically analyzing the correlation between the degree of damage and the degree of stress applied.

Our response – We thank Reviewer 1 for pointing out that our intention in Figure 4 might be somewhat ambiguous. In planning our figures, it was obvious that we needed to include some representative SEM images of micropillar deformation as that is a key feature of this experiment. In doing so with Figure 4, we chose to illustrate (as stated in the Figure 4 caption in line 409 and discussed more at length in lines 412-423) differences in mechanical outcome not only between micropillars of same geometry with different modulus, but also differences in the same sample at different locations along the path of the replica finger. We also chose to introduce these three samples in Figure 4 as case studies for further analysis later. We took this approach (i.e. three case studies that appear at various points in the paper) because it would be untenable to examine the deformation mechanics of all samples or even a significant fraction of the 69 permutations we tested. 

The three micropillar examples thus shown also appear as subjects of our mechanical analysis in Figure, 6 as well as in the S6 Appendix where they are treated to Finite Element Analysis. We were hopeful that our methodology would be more clear to the readers if we were treating micropillar samples with which they were already familiar. 

In regards to analyzing the correlation between the degree of damage and the degree of stress applied, we are grateful to Reviewer 1 for validating our struggle to reach the ultimate objective of our mechanical analysis! Energy accounting between the total applied tangential and normal stresses (that are both known) and the degree of deformation (that can be calculated for single micropillars) seemed obtainable at the onset of our analysis. It was the purpose for which we constructing a model of contact between the replica finger and the micropillar arrays. We thought that, since we were able to measure all the relevant geometrical parameters of the contacting bodies, we could model what we were unable to observe experimentally according to a hypothetical distribution of the applied forces. We even developed a computational model in MATLAB which was predicated on distributing these forces according to the geometry of fingerprint ridges incident on micropillars according to the density of the arrays in question. The output of this code was the deformation of single micropillars that would be subjected to this force distribution. 

Unfortunately, we were unable to achieve the goal of this analysis, largely because we were unable to reconcile this distribution of forces with observed results. In the Supplementary Information, the S5 Appendix (Additional Error Analysis on Contact Model Calculations) contains further examinations of this discrepancy and possible reasons why contact between the replica finger and the counter-surface was discontinuous. In short, the force applied to the system is not transferred very efficiently to the micropillars due to indeterminate losses in the apparatus as well as the viscoelastic dissipation of the replica finger itself. In addition, the evolution of contact area while the replica finger was in motion was unpredictable largely due to the indeterminate microroughness of the molded replica fingerprints. 

We feel that this Appendix has useful insights for readers wanting to understand the rough-on-rough contact of human fingerpads with relief structures or test programs intending to mimic the same. We chose to relegate it to the Supplementary Information because 1) as Reviewer 2 stated, this paper is already long enough and 2) we were unable to arrive at the desired agreement for a variety of reasons outlines in the text and SI. Given the immense amount of time and resources we have already devoted to addressing the reviewer’s comment (approximately 6 months of analysis that did make it into the submitted version of the main text), we respectfully submit that we allow this aspect of the project to be published as-is.

Reviewer 1, comment 4 – The authors used the finger replica (PDMS) when conducting the experiments. However, the Authors should analyze the similarities (e.g. strength, surface energy, and fingerprints) of the replica of the finger to replace a real finger.

Our response – We thank Reviewer 1 for again tendering a helpful suggestion that would increase the quality of our publication. While we touched upon some characteristics that our PDMS replica finger has in common with a real finger (such as modulus and fingerprint topology), we could have been more thorough in discussing differences such as what you mentioned. We brought up the hydration of real skin and its effects, both on the mechanical properties of a real finger as well as lubrication (in the Conclusion, lines 210-218, along with 770-776, reproduced below for convenience). 

“Upon release of the mold, we subjected the surface of the replica finger to a UV-ozone treatment to make the surface glassy and to reduce the tackiness.(49,50) This material could not however, simulate the extent that lubricated sliding occurs on real fingers due to lubrication from sweat, sebum and other oils, nor does it approximate possible osmotic repulsion due to the electric double layer that forms on the skin in atmosphere.(51) The contribution and variability of these factors in real fingers when compared to a more simplified contact model such as our replica PDMS finger could be an issue to investigate further. Regardless, we assumed that the contact probe we used to simulate interrogatory touch could be modeled as an elliptical region of skin-compliant elastomer with roughness imparted by periodic arrays of ridges.”

“Future work is needed to quantitatively understand the interaction of human fingers with micropatterned surfaces, and great care should be taken to select materials that are highly characterized, especially the viscoelastic material chosen to mimic skin. Human fingertip mechanical properties are also highly influenced by skin hydration(70) and while this would require further environmental controls, it would be insightful to investigate the effects of lubrication at the interface.”

In summary, it is true that the similarities and differences between our replica and a real finger is an important issue, and we have acknowledged these differences. Nevertheless, our study is the first which uses actual molded fingertips and realistic contact pressures, and probably represents the most accurate model for such fingertip tribology in the literature (short of using the fingertips of actual human participants, which introduces a range of convoluting variables).

Reviewer 2:

Reviewer 2, summary comment – In this work, the authors reported the collapse of polymer microstructures under external mechanical perturbations. As the authors described, high aspect ratio polymeric microstructures have a wide range applications in diverse engineering fields. Thus, the study on their mechanical stability is of great importance. Although such studies on the stability of polymeric microstructures have been extensively studied during the heyday of soft lithography, the authors performed more advanced, in-depth theoretical and experimental analyses by considering practical situations. The overall results and analyses are reasonable. Thus, I recommend this work to be accepted without significant alterations. The one minor comment is that the manuscript is quite long. I recommend that the authors consider if the length of the manuscript can be reduced for more conciseness of the manuscript.

Our response – We would like to thank Reviewer 2 for their encouragement! This study is the result of four years of careful work and a painstaking analysis of more than a thousand SEM images and detailed analysis of the relevant literature. We have spent much time both in analysis and in striving to communicate our findings to the best of our ability. We were very cognizant of the length of the manuscript and therefore designated a considerable amount of material for the SI. In fact, prior to submission, we moved several additional sections to the SI, so what was included in the main article is already an abbreviated version of our complete analysis.

---

## [Decision Letter · Decision Letter 1]

28 Jul 2021

Survival of Polymeric Microstructures Subjected to Interrogatory Touch

PONE-D-21-08649R1

Dear Dr. Lipomi,

We’re pleased to inform you that your manuscript has been judged scientifically suitable for publication and will be formally accepted for publication once it meets all outstanding technical requirements.

Kind regards,

Tae-il Kim

Academic Editor

PLOS ONE

Additional Editor Comments (optional):

Reviewers' comments:

Reviewer's Responses to Questions

**Comments to the Author**

1. If the authors have adequately addressed your comments raised in a previous round of review and you feel that this manuscript is now acceptable for publication, you may indicate that here to bypass the “Comments to the Author” section, enter your conflict of interest statement in the “Confidential to Editor” section, and submit your "Accept" recommendation.

Reviewer #1: All comments have been addressed

Reviewer #2: All comments have been addressed

2. Is the manuscript technically sound, and do the data support the conclusions?

Reviewer #1: Yes

Reviewer #2: Yes

3. Has the statistical analysis been performed appropriately and rigorously? 

Reviewer #1: Yes

Reviewer #2: Yes

4. Have the authors made all data underlying the findings in their manuscript fully available?

Reviewer #1: Yes

Reviewer #2: Yes

5. Is the manuscript presented in an intelligible fashion and written in standard English?

Reviewer #1: Yes

Reviewer #2: Yes

6. Review Comments to the Author

Reviewer #1: The Authors have well addressed the issues raised regarding the understanding of microstructure deformation. The authors provided proper explanations to support their arguments and the changes in the manuscript were appropriate. I recommend the paper could be published under PLOS ONE without further change.

Reviewer #2: The authors have well addressed reviewers' comments and I recommend the acceptance of this work.

7. PLOS authors have the option to publish the peer review history of their article (what does this mean?). If published, this will include your full peer review and any attached files.

Reviewer #1: No

Reviewer #2: No

---

## [Editor Report · Acceptance letter]

24 Aug 2021

PONE-D-21-08649R1 

Survival of polymeric microstructures subjected to interrogatory touch 

Dear Dr. Lipomi:

I'm pleased to inform you that your manuscript has been deemed suitable for publication in PLOS ONE. Congratulations! Your manuscript is now with our production department. 

Kind regards, 

on behalf of

Dr. Tae-il Kim 

Academic Editor

PLOS ONE